# The fitness of an introgressing haplotype changes over the course of divergence and depends on its size and genomic location

**Andrius J. Dagilis**⊙*, **Daniel R. Matute**

Biology Department, University of North Carolina, Chapel Hill, North Carolina, United States of America

* adagilis@email.unc.edu

**Data Availability Statement:** All data and code underlying figures presented here can be found in S1 File as well as a repository at https://github.com/adagilis/fitness_of_introgression.

## Abstract

The genomic era has made clear that introgression, or the movement of genetic material between species, is a common feature of evolution. Examples of both adaptive and deleterious introgression exist in a variety of systems. What is unclear is how the fitness of an introgressing haplotype changes as species diverge or as the size of the introgressing haplotype changes. In a simple model, we show that introgression may more easily occur into parts of the genome which have not diverged heavily from a common ancestor. The key insight is that alleles from a shared genetic background are likely to have positive epistatic interactions, increasing the fitness of a larger introgressing block. In regions of the genome where few existing substitutions are disrupted, this positive epistasis can be larger than incompatibilities with the recipient genome. Further, we show that early in the process of divergence, introgression of large haplotypes can be favored more than introgression of individual alleles. This model is consistent with observations of a positive relationship between recombination rate and introgression frequency across the genome; however, it generates several novel predictions. First, the model suggests that the relationship between recombination rate and introgression may not exist, or may be negative, in recently diverged species pairs. Furthermore, the model suggests that introgression that replaces existing derived variation will be more deleterious than introgression at sites carrying ancestral variants. These predictions are tested in an example of introgression in *Drosophila melanogaster*, with some support for both. Finally, the model provides a potential alternative explanation to asymmetry in the direction of introgression, with expectations of higher introgression from rapidly diverged populations into slowly evolving ones.

## Introduction

Before reproductive isolation is complete between 2 taxa, these taxa are often able to exchange migrants, form hybrids that backcross, and as a result exchange genes (for recent reviews, see Edelman and Mallet [1], Aguillon and colleagues [2]). Even a small number of hybrids between young species pairs can lead to introgression of genetic material between them [3,4], and studies across eukaryotes have identified introgression between a vast range of species [5]. Studies of hybrids have identified that the degree of divergence between the parental species strongly

**Funding:** This work was supported by the National Institute of General Medical Sciences of the National Institutes of Health (NIH) under Award R35GM148244 to DRM. AJD was supported under the National Institute of Allergy and Infectious Diseases of the National Institutes of Health (NIH) Award T32-AI052080. The funders had no role in study design, data collection and analysis, decision to publish, or preparation of the manuscript.

**Competing interests:** The authors have declared that no competing interests exist.

predicts hybrid fitness [6–9], and many models have considered how the fitness of these hybrids changes over the course of divergence, whether using insights from the build-up of incompatibilities [10–13] or through Fisher's Geometric Model [4,14–16]. Generally, these models find that there is a "gray zone" of speciation—a period in the process during which hybrid fitness is reduced yet remains high enough to enable gene flow between species as observed in real data [17]. There are many fewer models of introgression, and those that exist generally assume that introgression is primarily deleterious [3,18–23] despite the observation of adaptive introgression [1,24,25]. What types of loci and what fraction of the genome in general can introgress at different points in the speciation process remains unknown, but it is likely that introgressed regions are often non-neutral.

Several patterns are suggestive of the non-neutrality of introgressed regions. First, many introgressed alleles have been identified as adaptive or deleterious [1]. Further, in one of the few cases where historic data is available, the frequency of introgressed Neanderthal ancestry in humans has decreased over time [18]. Finally, correlation between recombination and introgression is suggestive of the non-neutrality of introgressed regions in a variety of systems. A positive relationship between recombination and introgression has been explained by "hybrid load," or the idea that alleles from a minor parent ancestry (the ancestry with less representation in the genome) are selected against [21,26]. The source for this selection is less clear—it is possible that the introgressing haplotypes carry weak deleterious epistatic interactions with the recipient genome (e.g., swordtail fish [27] or butterflies [28]), or that they are simply directly deleterious due to the accumulation of deleterious mutations in a more inbred population (e.g., Neanderthals [18,20,29]). In both cases, a clear negative correlation between the size of the introgressing haplotype and its fitness is expected—larger haplotypes contain both more deleterious alleles and neutral/adaptive alleles are selected out alongside these deleterious alleles in absence of recombination. High recombination regions can break such haplotypes up and enable neutral or adaptive alleles to introgress more easily, leading to a positive correlation between recombination rate and introgression. By contrast, a negative relationship between recombination and introgression has been observed in *Drosophila melanogaster* [30]. Recent work has used simulations to demonstrate that this relationship could be in part explained by positive selection on introgressed regions [31]. Empirical evidence of positive selection of introgressed alleles more generally is extensive (see Edelman and Mallet [1], Aguillon and colleagues [2] for reviews). Thus, introgressed alleles may commonly be both positively and negatively selected, but it is not clear how these effects may change over the course of divergence.

The selective effects of introgression have important theoretical consequences. On the one hand, alleles acting as barriers to gene flow (due to lower fitness in the recipient genome) may impede introgression of nearby neutral alleles [3] or even positively selected alleles [23]. If heterozygotes experience higher fitness due to overdominance of alleles, the introgression probability of neutral alleles may be increased [22], even when deleterious epistasis between an introgressed haplotype block and the recipient genome is frequent. If selection on introgressed alleles is positive, but distributed broadly across introgressing regions, adaptive blocks may be maintained at intermediate sizes due to linkage between many individually selected alleles [32]. In general, the probability that a given allele will introgress depends more heavily on its genetic neighbors than on its additive effects alone [33], and so understanding the overall fitness of a block of introgressing genes is a critical component for predicting patterns of introgression.

Introgressing haplotypes can carry a multitude of fitness effects. These effects will include the direct fitness effects of individual alleles as well as any potential incompatibilities with the recipient genome. However, introgressing haplotypes are likely to carry epistatic interactions

not only with the receiving genome, but also among the introgressing alleles as well. As those alleles arose in a common genomic background, epistatic interactions between them are tested by selection and are likely to be on average nonnegative. Thus, larger introgressing haplotypes carrying co-adapted sets of alleles may in fact be buffered against weak negative epistatic effects with the recipient genome. This within-block epistasis is particularly relevant when recombination cannot break up the introgressing haplotype, a scenario that is expected when recombination is absent (e.g., horizontal gene transfer in asexual lineages) or is regionally suppressed (e.g., sex limited chromosomes [34], inversions [25,35], or other recombination suppressors [36]). However, knowing the marginal fitness of a block that can recombine is also crucial in determining the long-term evolutionary dynamics of introgression. Currently, no theoretical approach has examined how the fitness of an introgressing haplotype changes depending on both its size and divergence between the 2 populations.

In this paper, we fill that gap. We begin by asking what selective forces act on different types of introgressing alleles—novel alleles that introduce derived variants at an ancestral site, ancestral alleles that introduce an ancestral variant at a derived site in the recipient population, and replacement alleles that introduce a different derived allele from the donor at a site in which the recipient population has fixed a derived allele. We then ask how the fitness of small haplotypes consisting of a small number of introgressing alleles changes as the total number of substitutions in the recipient population increases. We find that the overall divergence between species is a key predictor of both the direction and genomic location of introgression. In general, we expect introgression to be more strongly selected against when it replaces existing derived variation, because it breaks apart epistasis within the recipient population. More broadly, large haplotypes carrying many derived alleles may make it across species barriers with relative ease while divergence between populations is low, as the number of incompatibilities is smaller than the number of co-adapted interactions. As divergence continues, larger haplotypes are more and more strongly selected against, especially as they break up positive epistatic interactions within the recipient population. This model creates a new framework with which introgression over the course of speciation can be explained, provides several testable predictions, and suggests further avenues for both theoretical and empirical approaches. We examine some of these predictions in genomic data from populations of *D. melanogaster* which have experienced recent introgression [30,37] and find that introgression rarely occurs in regions of the genome in which the recipient population has evolved quickly in comparison to the source of introgression and occurs more in lower recombination regions.

## The model

We model a haplotype (a non-recombining block of the genome) introgressing from a source population (A) into a recipient population (B). We assume a total $b$ derived substitutions have been fixed in B since the population diverged from the common ancestor (Fig 1A). We ask what the marginal fitness of an individual carrying a rare introgressing haplotype is ($W_I$) compared to the average individual in the recipient population ($\bar{W}$), and calculate the effective selection coefficient ($\sigma_I$) for the haplotype, equal to $W_I/\bar{W}$ -1. We will consider both direct (individual) and pairwise epistatic fitness effects among all diverged alleles. Since we are setting fitness relative to the ancestor, all ancestral alleles have no epistatic effects with diverged alleles. We assume that fitness is a product of all relevant effects, but our results are congruent to assuming that fitness is a sum of these effects. We only examine the mean effect of alleles, assuming that variance in fitness effects is very low. We will first examine results in a haploid population, but they extend readily to diploid cases. First, let us examine the fitness of the recipient population (compared to the ancestor). This fitness will depend on $B$, the set of

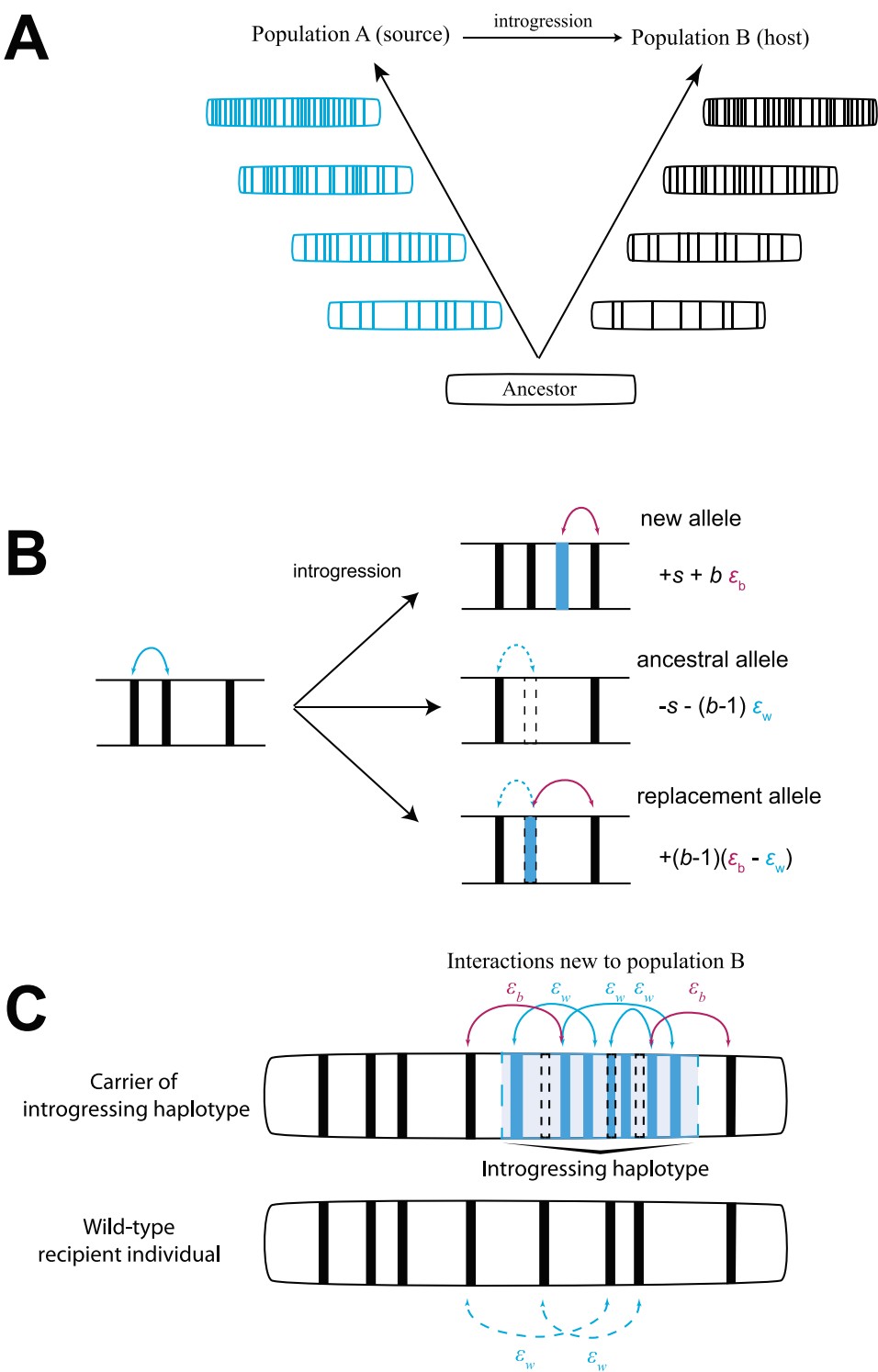

**Fig 1. Outline of model assumptions. (A)** Two populations (A and B) evolve independently, fixing substitutions in their genome, represented by filled bars, until introgression at some point moves alleles from A into B. **(B)** Three types of alleles may introgress as a result: new alleles which are derived in A and ancestral in B (blue fill), ancestral alleles which are ancestral in A and derived in B (dashed outline) and replacement alleles in which both A and B have fixed alternate variants (dashed outline, blue fill). Each of these types of alleles will on average have different fitness effects, and either introduce new epistatic interactions (solid arrows), or remove existing ones (dashed arrows) **(C)** An

introgressing haplotype of length $x$ carrying all 3 types of alleles will both introduce novel epistatic interactions (solid arrows) and remove existing ones (dashed) in individuals carrying the haplotype compared to others in the population.

substitutions fixed in that population, for a total of $b$ substitutions, and the direct and epistatic fitness effects:

$$\bar{W} = \prod_{i=1}^{b}(1 + s_i) \prod_{i,j \in B}(1 + \varepsilon_{ij}) \approx \prod_{i=1}^{b}(1 + s) \prod_{i,j \in B}(1 + \varepsilon_w) \approx 1 + bs + \binom{b}{2}\varepsilon_w. \quad (1)$$

Where $s$ is the average direct fitness effect of substitutions, and $\binom{b}{2}$ denotes the binomial coefficient "$b$ choose 2" equal to the number of pairs among $b$ total substitutions. Substitution $i$ carries a direct fitness effect $s_i$, and each pair of alleles $i,j$ has an epistatic effect of $\varepsilon_{ij}$. Finally, $\varepsilon_w$ denotes the average strength of epistatic interactions of alleles that have evolved "within" the same genetic background and $s$ denotes the average additive effect of fixed substitutions. The approximation on the right side of Eq (1) is accurate whenever the selective effects (both direct and epistatic) are very small and have low variance. The exact form of Eq (1) is used to calculate all results in figures, but the approximation is useful in describing how parameters impact fitness. For example, note that fitness in the recipient population depends linearly on the number of substitutions, but the number of pairwise epistatic interactions increases quadratically.

## Selection on individual alleles

We can now consider introgression of 3 different classes of alleles (Fig 1B). The first is a site at which a substitution occurred in population A but not in B, termed "new" alleles. The selection coefficient on this allele can be expressed as:

$$\sigma_{new} = \frac{W_I}{\bar{W}} - 1 = \frac{b(1 + \varepsilon_b) \prod_{i=1}^{b+1}(1 + s) \prod_{i,j \in B}(1 + \varepsilon_w)}{\bar{W}} - 1 \approx s + b\varepsilon_b \quad (2)$$

That is, the allele carries 1 new direct fitness effect, as well as $b$ epistatic interactions of strength $\varepsilon_b$—epistasis of alleles "between" populations. Unlike within population epistatic interactions, these will be epistatic interactions that have not been tested by selection (as they are between alleles evolved in different backgrounds) and so are likely to be quite different (see Dagilis and colleagues [13] for simulations confirming this intuition). Indeed, while $\varepsilon_w$ can on average be expected to be positive, $\varepsilon_b$ is likely to be both negative and an order of magnitude weaker. What these parameters suggest is that even alleles with directly beneficial fitness effects will be selected against as populations diverge ($b$ increases). On the other hand, if epistasis between populations is positive for some reason (due to the shape of the fitness landscape, for example), introgression of these alleles will on average be selected for.

A second type of allele that can introgress is what we will call an ancestral allele—a position in the genome in which the recipient population has fixed a substitution, but the introgressing allele carries an ancestral variant. Following the logic from above, the selection coefficient of an allele of this type can be expressed as:

$$\sigma_{ancestral} \approx -s - (b - 1)\varepsilon_w. \quad (3)$$

In this case, the introgression of the ancestral allele removes a direct fitness effect as well as $b$-1 epistatic effects with existing substitutions in the recipient population. Thus, it is immediately clear that fitness effects of introgressing ancestral variation will in general be more

deleterious than those of introgressing new alleles under reasonable parameter space (positive within population epistasis), since they remove whatever the direct fitness effect is and remove positive epistasis. As with the introgression of new alleles, the total selection on these alleles increases as the recipient population becomes more diverged (larger $b$).

Finally, we can consider what occurs when an introgressing allele is derived compared to the ancestral allele, but also replaces an existing derived allele in population B. In this case, it is reasonable to assume that the 2 fitness effects are essentially combined—we can call this type of allele a "replacement" allele and express its selection coefficient as:

$$\sigma_{replacement} \approx (b-1)(\varepsilon_b - \varepsilon_w) \tag{4}$$

Here again, whenever $\varepsilon_b < 0 \; and \; \varepsilon_w > 0$, increased divergence leads to stronger selection against introgression; however, direct selection coefficients do not play a role as we assume direct effects of alleles fixed in either population are equivalent. Since epistasis may often be orders of magnitude weaker than direct fitness effects, these types of alleles may be very weakly selected while $b$ is small (early in divergence). Note that in the special case $\varepsilon_b = \varepsilon_w$, no selection against these types of alleles is expected under this model.

## Selection on haplotypes

Introgression of a larger block of alleles is likely to contain loci of all 3 types (Fig 1C). When more than a single allele introgresses, the effect of the full haplotype is not simply a sum of the effects of Eqs (2) to (4), but also includes effects between the alleles on the haplotype itself. The only new type of interaction will be within population epistatic interactions ($\varepsilon_w$) between alleles coming from population A (either new or replacement alleles). That is, each pair of those alleles has its own small epistatic interaction of strength $\varepsilon_w$.

To simplify notation somewhat, we can consider that the introgressing haplotype will carry alleles at 2 types of sites—ones in which the source population (A) has fixed substitutions, and ones in which the recipient population (B) has fixed substitutions. New alleles will be of the first type, while ancestral of the second, and replacement alleles are of both types. Letting the numbers of these be $x_A$ and $x_B$, respectively, we define the size of the haplotype ($x$) as the total number of alleles it is carrying, which will be equal to the number of new, ancestral, and replacement sites. Since sites with derived alleles fixed in A can be either new or replacement alleles, $x_A = x_{new} + x_{replacement}$ and $x_B = x_{ancestral} + x_{replacement}$. With all of these assumptions, the effective selection coefficient of an introgressing haplotype in a haploid population can be expressed as:

$$\sigma_I \approx s(x_A - x_B) + \frac{\varepsilon_w}{2}\left(x_A^2 - x_A - x_B(2b - x_B - 1)\right) + \varepsilon_b x_A (b - x_B) \tag{5}$$

We can further simplify the above relationship by assuming that replacement alleles will be quite rare early in the process of divergence, and letting the fraction of $x$ composed of ancestral alleles be denoted as $f$ with the remainder being new alleles (e.g., $fx = x_{ancestral}$, $(1-f)x = x_{new}$). The selection coefficient on an introgressing haplotype can then be expressed as:

$$\sigma_I \approx (1 - 2f)\left(sx + \frac{\varepsilon_w}{2}(x^2 - x)\right) - x(\varepsilon_w f - \varepsilon_b(1 - f))(b - xf) \tag{6}$$

Note that the number of fixed variants in the recipient population ($b$) only appears as a linear term, while the size of the introgressing haplotype ($x$) appears as a quadratic term. As a result, the size of the haplotype can have a larger effect on its fitness than the divergence of the recipient population while $b$ is of a similar order of magnitude to $x$. We can make 2 final

simplifications for early and late divergence. Early in divergence, if a haplotype only carries newly derived alleles, Eq (6) further simplifies to:

$$\sigma_I \approx sx + \varepsilon_w \binom{x}{2} + \varepsilon_b xb \qquad (7)$$

As divergence continues, and $b >> 1$, we can instead assume that all alleles on the introgressing haplotype will be replacement alleles. When $x << b$, the total fitness of the haplotype can then be approximated as:

$$\sigma_I \approx xb\varepsilon_b \qquad (8)$$

The replacement alleles introduce as many within population epistatic interactions as they remove, leading to a linear relationship between haplotype size and its fitness.

We can parameterize the exact form of Eq (5) to examine the exact effects of $b$ and x. The exact values of the relevant parameters are hard to gauge—very few studies of epistatic effects between many alleles have been performed (but see Costanzo and colleagues [38]). In general, some large differences between $\varepsilon_w$ and $\varepsilon_b$ may be expected. Within-population interactions are exposed to selection, and are therefore not only likely to be positive, but in simulations have been shown to be an order of magnitude larger than between-population interactions [13]. On the other hand, because between population interactions are not exposed to selection, they are likely to be a random subset of interactions among all possible epistatic interactions. For illustrative purposes, we will assume that $\varepsilon_b = -10^{-4}$ and that $\varepsilon_w$ is an order of magnitude larger and positive in line with prior simulations [13]. Nonetheless, these simulations were based on a network of interactions between knockout mutations and so may be orders of magnitude larger than most epistatic interactions. We examine a slightly larger parameter space in S1 Fig, but we try to draw conclusions from the general shapes of the fitness curves rather than their exact values. Finally, we chose a value of $s$ (−0.01) that is larger than any of the epistatic effects—the actual fitness landscape may be highly different (and indeed all these values may change with divergence), so caution must be employed in interpreting the actual values of the curves. As a result, we will talk about "small" and "large" haplotypes, as well as "early" and "late" divergence, but the exact values of small or early depend entirely on the values of the selection parameters in a system.

When divergence is low (and by consequence $b$ is small), introgression of large haplotypes carrying new alleles is strongly favored (Fig 2A). As divergence continues, the size of the introgressing haplotype necessary to overcome incompatibilities becomes larger, and even small haplotypes are strongly selected against (Fig 2A). At some degree of divergence, the number of alleles necessary to overcome the number of incompatibilities becomes so large as to be practically impossible, leading to a purely negative relationship between the size of the haplotype and fitness over feasible haplotype sizes. A second effect of divergence is that haplotypes become increasingly unlikely to carry solely new alleles as the region they introgress in is more and more likely to have fixed substitutions in the recipient population. Replacing existing variants with either ancestral alleles or new derived alleles decreases the fitness of the introgressing haplotype strongly (Figs 2B and 3). This is the case even though we assume the direct effects of the fixed substitutions are deleterious (i.e., removing each diverged allele without accounting for epistatic effects would be beneficial). If we further expand to vary the number of new, ancestral, and replacement alleles at the same time, we find that replacement alleles act largely like ancestral alleles, especially later in divergence (Fig 3)—they are generally deleterious because they break up large numbers of existing co-adapted sets of alleles, and while they bring in some positive epistasis among each other, the number of those interactions is relatively small in comparison. While we choose very large epistatic effects here for

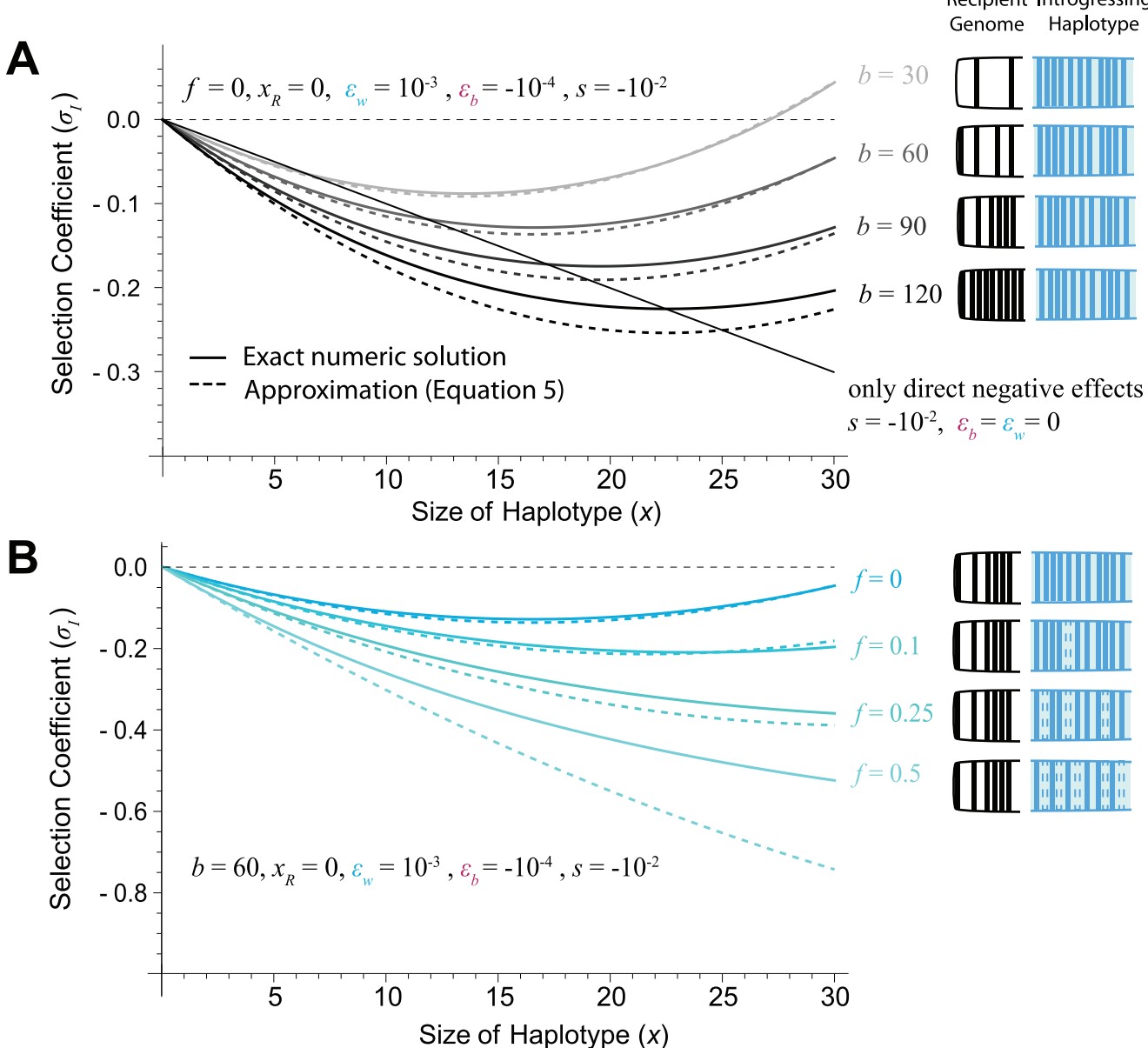

**Fig 2. Results for a haploid model. (A)** When divergence ($b$) is low, even fairly small introgressing haplotypes carry sufficient positive interactions between introgressed derived alleles to cancel out potential deleterious effects. As divergence increases, however, the fitness of smaller haplotypes rapidly declines, and it takes increasingly larger haplotypes to cancel out potential DMIs. If each allele has only direct negative selection on it in the receiving population (solid black line), a simple linear relationship in fitness is expected. Note that we assume direct selection on each allele is on average negative, and so the positive selection is entirely driven by epistasis. **(B)** When the introgressing haplotype is only carrying novel alleles ($f = 0$), increasing the number of alleles introgressing introduces increasing amounts of positive epistasis. However, as it replaces more and more of existing substitutions in the receiving population, it becomes more and more strongly selected against. In both panels, it is assumed all alleles are carrying either new or ancestral variants, with no replacement alleles ($x_{replacement} = 0$). Solid lines show exact numeric solutions, while dashed lines show approximations from Eq (5). The code underlying this figure can be located in S1 File.

illustrative purposes, they are based on observed epistatic effects in yeast mutants [38], and so may represent reasonable values for the introgression of haplotypes carrying large effect alleles. Smaller epistatic effects are likely to change the scale of divergence needed to see selection against introgressing haplotypes as well as the size of the introgressing block before

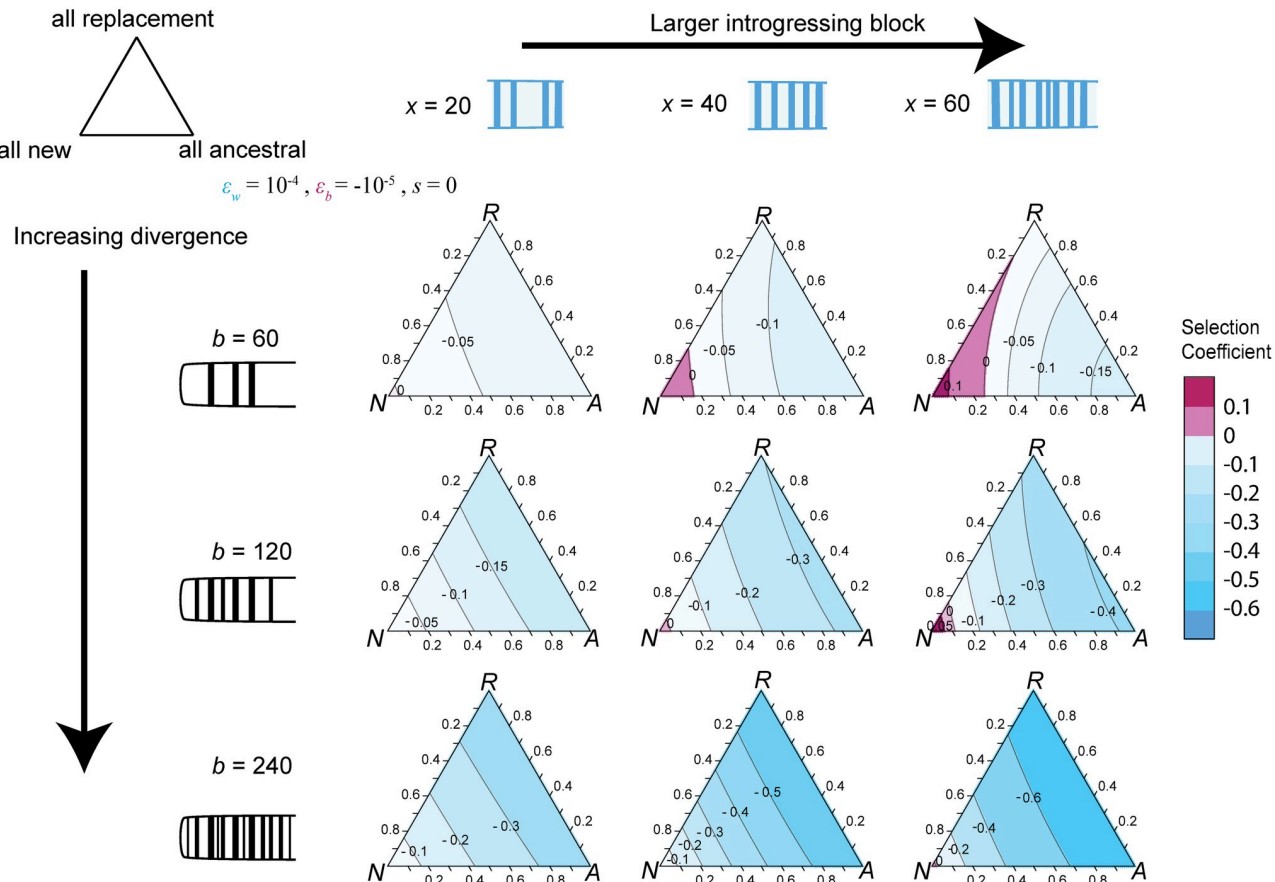

**Fig 3. The effect of replacement alleles mimics introgressing ancestral variation.** Examining how the selection coefficient on an introgressing haplotype changes as the fraction of substitutions it carries varies between new, ancestral, and replacement alleles. Parameters listed in top left and calculations based on exact numerical evaluation of fitness (see S3 Fig for approximation using Eq (5)). Early in divergence (top row), carrying more ancestral variants has a more strongly deleterious effect than carrying replacement alleles. However, as divergence increases (lower plots), haplotypes that carry replacement alleles begin looking largely like haplotypes that carry entirely ancestral variants. In all cases, haplotypes carrying only novel alleles are the least deleterious. The code underlying this figure can be located in S1 File.

it becomes positively selected. We can predict both of these quantities by examining the inflection point of Eq (6).

Since $\sigma_I = 0$ when $x = 0$ (by definition, as there is no selection on a haplotype carrying no alleles), the quadratic shape of the fitness function (Eq (6)) and reasonable parameter assumptions ($\varepsilon_w > 0 > \varepsilon_b$) suggest there is a minimum fitness of an introgressing haplotype dependent on its size (S2 Fig). This minimum will occur when the derivative of Eq (6) with respect to x is 0. If we ignore direct fitness effects (i.e., let $s = 0$), this value occurs at:

$$b\frac{\varepsilon_w f - \varepsilon_b(1-f)}{\varepsilon_w - (\varepsilon_w + \varepsilon_b)2f(1-f)} \tag{9}$$

When $f = 0$ (i.e., only new alleles are being introduced), this is simply $-b\frac{\varepsilon_b}{\varepsilon_w}$, so it depends entirely on the ratio of between and within-population epistasis (S2 Fig) and divergence of the recipient population. If between-population epistasis is several orders of magnitude weaker than within, the most strongly selected-against haplotypes will be a small fraction of the divergence in the recipient population. For example, if $\varepsilon_b \approx 10^{-6}$ and $\varepsilon_w \approx 10^{-4}$, then the minimum

fitness of an introgressing haplotype occurs when it introduces $0.01b$ new alleles. Any larger haplotypes will be increasingly less deleterious and potentially positively selected. When both $\varepsilon_b$ and $\varepsilon_w$ are positive, this inflection point is negative, meaning that all introgression is positively selected. Finally, if both epistatic parameters are negative, the inflection point is again negative, but all introgression is selected against. As within and between epistasis become more similar, this value gets increasingly closer to $b$, meaning unrealistically larger haplotypes are necessary to overcome any deleterious effects—e.g., if the recipient population has fixed 200 substitutions and $\varepsilon_b = -\varepsilon_w$, then haplotypes introducing up to 200 alleles will be increasingly selected against, equivalent to swamping the recipient genome with as many diverged alleles as it has already fixed. When $f = 1/2$ (so half of the alleles are replacing existing variants), the above is always equal to $b$, meaning that haplotypes are increasingly selected against until they replace more than half of the existing substitutions, and introduce an equal number of new alleles. Finally, when $f = 1$, Eq (9) is again equal to $b$, meaning that haplotypes that are only replacing existing variants are always more strongly selected against as they become larger (S2 Fig). Note that in this final case, the maximum value of $x$ is $b$, as if each allele is swapping a diverged variant for an ancestral one, then you can at most replace $b$ alleles.

## Selection in diploids

While results are straightforward for the haploid model, several new parameters need to be introduced for a diploid model (S4 Fig). Direct fitness effects are assumed to have dominance $h$. When alleles tend to be recessive ($h < 0.5$), the removal of existing variants has larger fitness effects, making introgression harder when more alleles are being replaced ($f > 0$) (S5 Fig). On the other hand, dominant alleles heighten the effect of introducing new variants, making introgression more difficult if alleles are negatively selected and only new alleles are introduced (S5 Fig). Epistatic interactions may also have varying degrees of dominance (S4 and S5 Figs). In particular, epistatic interactions may occur at different strengths depending on whether both derived alleles are heterozygous, one is heterozygous and the other is homozygous, or both are homozygous [12,13]. We parameterize this by setting the interaction strength when both alleles are homozygous as $\varepsilon$, while homozygous–heterozygous interactions are assumed to be on average of strength $a_2\varepsilon$ and heterozygous–heterozygous interactions are of strength $a_1\varepsilon$. A closed form expression for the approximate fitness of a haplotype introgressing in a diploid population can be found in the Materials and methods (Eqs [13,13]), but the addition of 2 extra parameters makes simple analytical conclusions more difficult to draw. Assuming that $a_1$ and $a_2$ are roughly proportional (e.g., both recessive or both dominant), several broad patterns can be identified. As epistasis becomes recessive, the role of new alleles is strongly weakened (S5 Fig). Especially early in the process of introgression, the introgressing haplotype is almost always going to be found in heterozygotes. As a result, any within population epistasis will be strongly dampened, while any between population epistasis will be quite weak (S5 Fig). This means that any new epistatic interactions the haplotype introduces have a very weak effect on fitness, while the removal of existing interactions plays a much stronger role. On the other hand, dominant epistasis would suggest that within effects are expressed nearly as strongly as in the haploid case, and between effects are immediately exposed to selection (S5 Fig). In our general case of positive within and negative between this leads to strong selection against the introgressing haplotype. If epistasis tends to be dominant, therefore, introgression should on average be more positively selected than if it is recessive. In nature, of course, introgression is likely to vary in dominance broadly both within and between systems.

## Effects of recombination

So far, we have discussed the model in terms of haplotypes unaffected by recombination. This simplifying assumption allows us to infer the effective selection coefficient of a set of alleles, but is not likely to be held in a variety of cases of introgression. We were unable to extend the results to an arbitrary linkage map; however, we can make some conclusions about introgression across the genome based on the results from the fully linked haplotype approach.

Imagine the genome is split up into haplotypes, the sizes of which are inversely correlated to local recombination rates. In turn, we would expect regions of the genome with high recombination to have small introgressing haplotypes, while regions of the genome with low recombination may carry much larger introgressing blocks. To further simplify, we will assume that any individual is at most carrying a single region with introgressed alleles. While the average fitness in the population (Eq (1)) will change with many introgressing haplotypes of various sizes, the relative fitness of each haplotype should still show the same shape, since average population fitness only appears in the denominator (i.e., the change is only quantitative, but not qualitative). As a result, we can make some predictions—early in the process of divergence, large introgressing blocks (found most likely in regions of low recombination) will have higher fitness than small ones (found in regions of high recombination), with intermediate blocks showing the lowest fitness. Thus, there may be a negative relationship between local recombination rate and introgression. Later in divergence, large haplotypes are less fit than small ones, and so a positive relationship between recombination rate and introgression is expected to occur. The latter pattern has been seen in many studies of introgression between species [26,39,40], while a negative pattern between recombination and introgression was observed in gene flow between populations of *D. melanogaster* [30].

## Introgression in *Drosophila melanogaster*

Within Africa, *D. melanogaster* is strongly genetically structured, with multiple distinct ancestries geographically distributed throughout the region [37]. Subsequent gene flow between these ancestries has been detected in multiple studies [30,37]. A particularly intriguing case is introgression of African ancestry into North American populations, in which a negative relationship between recombination rate and introgressed ancestry was previously identified [30,31]. In previous work, we identified recent introgression between an ancestry primarily present in Western Africa (called West) and an Out of Africa ancestry primarily present among North American flies (referred to as OOA2), confirming prior observations by Pool and colleagues. This introgression event represents an opportunity to test some of the predictions of our model for recently diverged populations. The ancestries in question likely have been separated for 10,000 years [41], and introgression is thought to be a result of the Triangle trade, taking place from the XVIth to the XIXth centuries. In previous work, we calculated a measure of introgression proportion within local genomic windows ($f_D$) as well as population differentiation ($F_{ST}$) in non-overlapping 200 SNP windows across the genome for these populations (Coughlan and colleagues, 2021). Here, we use those data to test several predictions of our model.

We first calculated the Population Branch Statistic (PBS) [42] for each of 3 populations: OOA2, West, and South1—the latter being an ancestry present in Southern Africa with no evidence of introgression with either of the other 2 populations. Briefly, PBS indicates what proportion of the total branch length for a window leads to the focal population. As introgression would lead to shorter branch lengths between the introgressing populations, we used $F_{ST}$ calculated excluding any individuals that carried more than 1% introgressed ancestry according to ancestry estimates performed using PCAngsd [43]. This gives us a baseline of divergence

between these ancestries without the effect of introgression. We next set all negative PBS values to 0, as a negative value of PBS is not intuitive to interpret (however, our results hold even when these windows are retained, S6 Fig). Finally, we re-normalize each window by dividing each PBS value by the total of the 3 PBS values at that window. The resulting values naturally sum to 1, letting us ask what proportion of evolution has likely occurred on each branch among the 3 populations. A West branch value of 1 for a window would therefore indicate that all substitutions within that window have taken place along the branch to the Western population, with South1 and OOA2 populations sharing all alleles. We next examined whether windows with high $f_D$ differed in their branch lengths compared to the genome average (Fig 4B). We find that while most of the genome has large OOA2 branch values, high introgression windows tended to cluster at higher South1 values. This recent introgression event seems to confirm some of the predictions made in our model, as these populations with low divergence show overall a negative relationship between introgression and recombination (Fig 4C) as

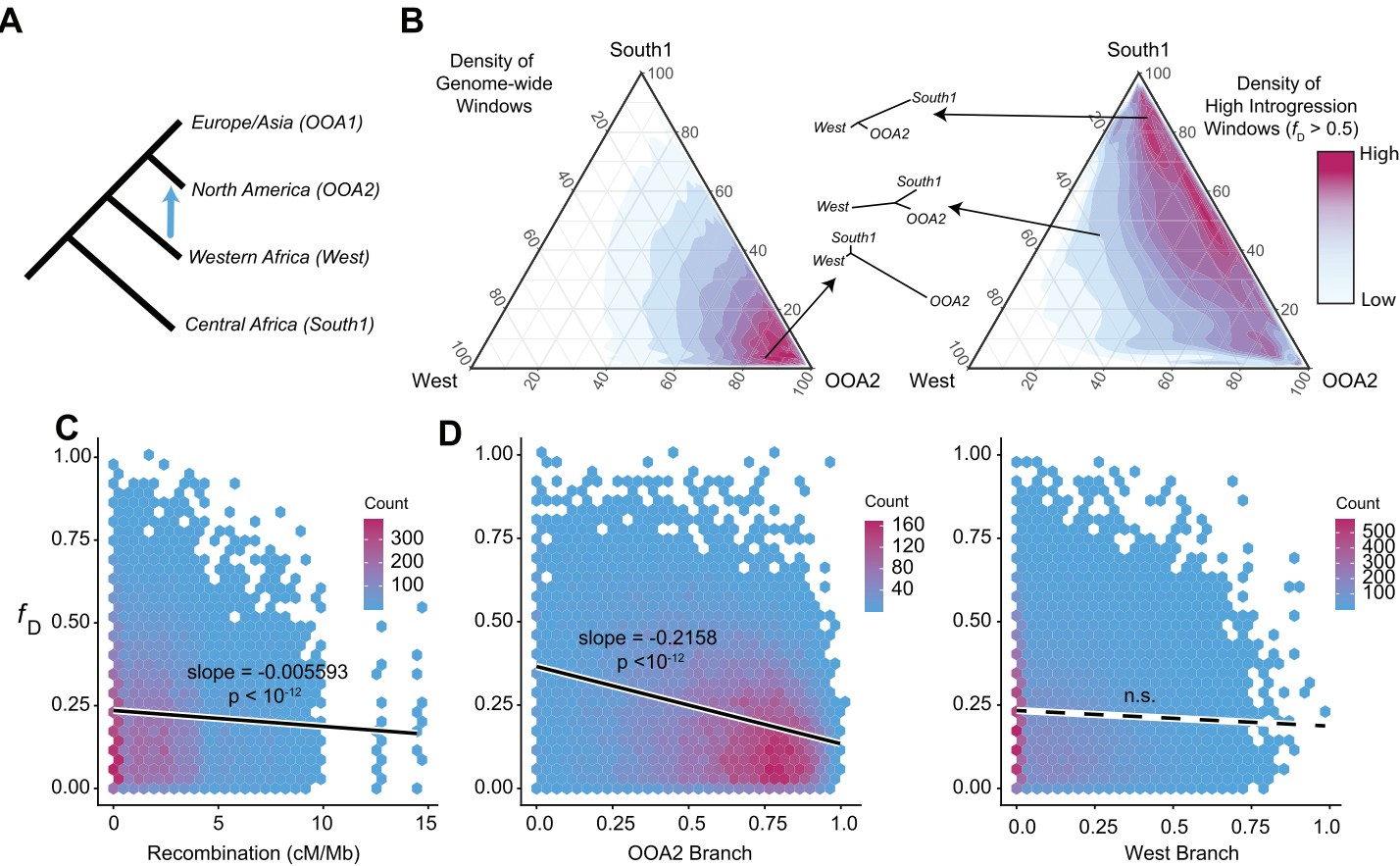

**Fig 4. (A)** Previous work has identified an introgression event between Western African *D. melanogaster* (West) into North American populations (OOA2). South African populations (South1) are ancestral to both and used as an outgroup. Using previously published data, we calculated branch scores (normalized Population Branch Statistics), for 25,874 windows across the genome with some evidence for introgression. $f_D$ statistics were calculated using the population relationship shown in A, with OOA1 as a sister population with OOA2. **(B)** The genome-wide density of relative branch scores is clustered at values indicating high differentiation between OOA2 and both of the other populations (red values—high density, blue—low). However, the higher introgression windows ($f_D > 0.5$) are depleted for long OOA2 branches and show many more windows with longer West and South1 branches. **(C)** Density plot of recombination rate versus $f_D$, each bin is colored by the number of windows within the bin. A negative relationship between local recombination rate and introgression exists for all windows with $f_D > 0$ and **(D)** a similar negative relationship exists for OOA2 branch score and introgression, indicating diverged windows are more resilient to introgression. No significant relationship between West branch length and $f_D$ was identified. Slopes and *p*-values are from individual linear models for each plot. The code and data underlying this figure can be located in S1 File.

observed previously [30,31]. Windows that have shown higher divergence in the recipient population show overall resilience to introgression (Fig 4D), with both relationships significant in linear models of the data. However, as recombination rate and PBS seem to be somewhat co-linear, we cannot easily separate out the causal relationship, and their co-linearity makes statistical analysis difficult. We do, however, find no significant relationship between the West branch and $f_D$. Note that because the relative branch scores are calculated using individuals with no admixed ancestry according to *NGSAdmix* [44], these patterns should not be driven by introgression reducing PBS scores for the OOA2 population.

## Discussion

Our model describes the fitness of an introgressing haplotype over the course of divergence. Divergence, the fraction of the haplotype that replaces locally diverged alleles, and (in case of diploidy) epistatic dominance play the largest roles in determining whether and what kinds of haplotypes can introgress. We discuss each of these in turn.

### Divergence

Early in divergence, when the recipient genome has fixed few substitutions, introgression is unlikely to replace new derived alleles. In turn, the majority of the alleles introgressing may be bringing in entirely new variants. While their exact fitness depends on how divergence between the 2 populations has taken place, these alleles are likely to be neutral or positively selected to have been fixed in the source population. While these alleles may have negative epistatic interactions with alleles fixed in the recipient population, there are relatively few of these possible incompatibilities [10,12]. Additionally, there is likely positive epistasis between alleles the introgressing haplotype is carrying [13]. The number of deleterious interactions with alleles in the recipient population increases linearly with the size of the introgressing haplotype, while the number of positive epistatic interactions increases quadratically (Eq (6)). In effect, larger introgressing haplotypes are expected to, on average, be more positively selected during the early stages of divergence. Even for alleles that are highly deleterious in the recipient population, positive epistasis may allow larger haplotypes to introgress, although this may only occur for haplotypes carrying very many alleles under some fitness landscapes (Fig 2A). The same is largely true for haplotypes introgressing in a diploid population; however, the dominance of epistasis plays a large role in determining selection on the introgressing haplotype (S5 Fig). Gene flow between populations with low levels of divergence is not often dissected in the same way as gene flow between diverged species, but several studies do suggest that, in general, the fraction of the genome which shows evidence of introgression decreases with increasing divergence [45]. Whether larger haplotypes are also introgressing at lower degrees of divergence is, as far as we are aware, unknown, but this pattern should in part result in a negative relationship between recombination rate and introgression, which has been observed in introgression between populations of *D. melanogaster* [30,31], and is repeated in this study (Fig 4C).

As species continue to diverge (and *b* increases), there is a buildup of reproductive incompatibilities leading to increased selection against introgressing haplotypes. In turn, it takes larger introgressing haplotypes to overcome the deleterious effects with the recipient population, with the inflection point defined in Eq (8) relating directly to total divergence. The number of alleles required to overcome incompatibilities with the recipient genome quickly becomes so large as to be implausible, leading to an increasingly negative relationship between the size of the haplotype and its fitness (Fig 2A). Therefore, introgression of smaller haplotypes, which are on average more weakly selected against, is much more likely than

introgression of large haplotypes. This relationship should manifest in a pattern of decreased introgression in regions of the genome with low recombination, a pattern observed in several species, for example, swordtails [26]. These results are also congruent with previous models that assume that larger introgressing haplotypes are broken down by recombination to avoid direct negative fitness effects [18] or to break up negative epistatic interactions [40].

Finally, divergence is also likely to play a role in modifying the other parameters of our model. As populations diverge, the relevant parameters of alleles fixed within each population ($s$ and $\varepsilon_w$) may change [13]. Mutations that are increasingly deleterious in the ancestral background may be able to fix due to a buildup of positive epistasis with previously fixed substitutions. In the figures presented in this manuscript, we assumed that direct effects would be an order of magnitude more deleterious than epistatic. Early in divergence, this may not be the case—it is unlikely that the first mutations to fix would be highly deleterious in an ancestral background. Even small haplotypes may be favored to introgress early in divergence (and indeed, individual alleles might be favored to introgress when divergence is extremely low). As divergence continues, $s$ may decline while $\varepsilon_w$ increases—making introgression increasingly more difficult. Similarly, $\varepsilon_b$ can change over the course of divergence as well, with its value determined entirely by the fitness landscape and where on it the 2 populations evolve to. Lastly, the fraction of alleles that replace local variants will naturally change over the course of divergence. While a haplotype is unlikely to replace fixed differences in the recipient population early in divergence, it becomes more likely to do so as divergence increases. Many of the factors crucial to determining the marginal fitness of an introgressing allele in Eq (6) may therefore change with respect to $b$, but our model is still useful in capturing the fitness of introgressing haplotypes given a fixed set of selective parameters.

## Fraction of ancestral and replacement alleles

Not only is an introgressing haplotype bringing in novel alleles, which now have many potential incompatibility partners, but also that haplotype is likely to replace existing derived alleles in the recipient population (i.e., see [46]). Whether this replacement occurs in the form of reintroducing ancestral variants or different diverged alleles (Fig 1B), larger haplotypes replace more diverged alleles, disrupting existing epistasis in the population. As we note earlier, the positive epistasis within a population has evolved under a selective sieve, and so it is likely to both be on average positive and of a larger magnitude than random new interactions between alleles untested in the same genetic background [13]. Because of this, as larger haplotypes replace more local alleles, they remove increasing numbers of positive interactions and may be strongly selected against (Fig 1C). This effect is paralleled in diploid populations (S5 Fig), although it depends strongly on the dominance of epistatic interactions —if epistatic interactions are dominant, a rare introgressing haplotype will not diminish the effects of within population epistasis very strongly. Whether the introduced alleles are simply bringing in ancestral variation or replacing existing derived variants with new ones, haplotypes carrying many such alleles are generally more strongly selected against than ones carrying derived variants at sites with ancestral variation in the recipient population (Fig 3). This means that introgression is expected to be strongly resisted in parts of the genome that have large numbers of diverged alleles, but not necessarily in regions carrying primarily ancestral variants. We find this pattern among populations of *D. melanogaster*, with a negative relationship between introgression and divergence of the recipient population (OOA2) from the source (West) (Fig 4D).

## Dominance of epistasis

In the haploid case, we can infer some reasonable parameter values (specifically for epistasis within and between populations) to make some general conclusions. Such conclusions are difficult to extend to diploids without a clear understanding of the dominance of epistatic interactions. While rare, the introgressing haplotype will occur primarily in heterozygotes. As such, when epistasis is recessive, introgressing haplotypes will not benefit from positive interactions among the alleles they carry, but replacement alleles will strongly disrupt existing positive epistasis and lead to overall stronger selection against introgression (S5 Fig). On the other hand, if epistasis is dominant, introgressing haplotypes do not disrupt existing interactions, and will bring in stronger positive effects among the alleles they carry, leading to much stronger selection for introgressed haplotypes of any size. Little is known about the dominance of epistatic interactions, but they may be recessive as suggested by studies of the large-X effect [12] and Haldane's Rule [47]. As with any biological system, actual substitutions are likely to have epistatic interactions of various dominance and so models of a single parameter for dominance are unlikely to fully represent reality. Nonetheless, the general shape of the fitness functions does not change based on the dominance parameters examined here, and so most of our predictions should hold in a variety of circumstances.

## Caveats

Our model is simplistic by design, in order to give a clearer intuition of the relationship between the size of an introgressing haplotype and its fitness during the course of divergence. To describe this relationship, we make several major assumptions. First, we assume that the haplotype has appeared in the recipient population, but do not know how. In order for a small haplotype to occur in the recipient population, many generations of backcrossing must first occur. Later in divergence, when selection against large haplotypes is very strong, it may be exceedingly rare for these small haplotypes to make it across species barriers, primarily because early generation backcrosses are strongly selected against (see [48] for a recent review). Indeed, we do not account for the fitness of F1 or later generation hybrids at all and simply measure the fitness of the haplotype against an otherwise uniform genetic background.

Second, we ignore recombination in our model, which prompts several important qualifiers. The haplotype's fitness determines its fixation only if that haplotype stays the same size—in practice, the fitness of a haplotype and its fixation probability are not identical. Haplotypes spanning large sections of chromosomes and carrying many alleles may be positively selected but are also more likely to be broken up by recombination. As noted in Sachdeva and Barton [33] "the initial introgression of loci within a block is governed by the rate of the spread of the block (rather than the selective effects of individual loci)," with block sizes being determined by recombination. In effect, while the fitness we calculate in our model may be an important factor, how recombination decays larger haplotypes into stable blocks plays a crucial role in the establishment of the introgressed region. In a simplified view, if we extend the logic of Sachdeva and Barton to our model, intermediate sized blocks may introgress most easily early in divergence, even if they are not the most fit. Our model is more accurate if the haplotype is being maintained at its size by an inversion or some recombination modifier linked to the haplotype, acting as a "supergene" [49–52], but most introgressing regions are unlikely to be maintained in perfect linkage disequilibrium. Similarly, because we ignore recombination, we equate the numbers of alleles it carries with its physical size. However, in practice, physically small regions may carry many more alleles than large ones, especially as gene density can be highly heterogeneous. As a result, our expectation of different relationships between

recombination rate and introgression probability depends in large part on an assumption of a roughly equal density of substitutions across the genome.

Lastly, we assume that there are only 2 genotypes in the population—individuals carrying the "recipient" genetic background and those carrying the introgressing haplotype. Even if we ignore polymorphism within the recipient population before hybridization, which may be highly relevant to fitness of hybrids in general [53,54], the 2 processes above—backcrossing and recombination—will lead to a wide array of genotypes in the recipient population. Real instances of introgression have the haplotypes evolving alongside many other introgressing haplotypes of various sizes and distributions along the genome. Early in divergence, this may dampen the benefits of keeping large blocks intact, as individuals may carry many unlinked co-adapted introgressing alleles that weaken selection against any individual introgressing allele. More broadly, this means that multiple haplotypes of varying sizes are likely to be introgressing, making predictions far less straightforward [21,22,31].

## Conclusions

Selection for or against introgressed haplotypes seems to follow similar patterns to fitness of hybrids in general. Early in divergence, hybrid fitness can remain high [13,15,16], and similarly we find that introgressing haplotypes may be positively selected early in divergence. Under reasonable parameters, co-adapted epistatic interactions can outweigh deleterious epistasis and lead to positive selection of large introgressing haplotypes. This is especially the case when the haplotype carries alleles at sites that have not experienced substitutions in the recipient population. On the other hand, later in divergence, as hybrid fitness decreases, haplotypes are negatively selected unless they are bringing in vast numbers of novel alleles, which is unlikely to occur in nature. Therefore, different relationships between introgressing haplotype size and fitness are expected between recently diverged populations and highly diverged species pairs.

There are 2 consequences from these simple observations. Late in divergence, recombination that breaks up introgressing haplotypes is nearly always favored, as smaller haplotypes have lesser fitness consequences. This expectation may explain the observed positive correlation between introgression and recombination in several systems, including swordtail fish, *Heliconius* butterflies, and Neanderthal-human introgression [26,55]. However, early in divergence, larger haplotypes might actually introgress more easily, although the degree to which they do so depends largely on the recombination map [32]. In turn, the relationship between recombination rate and introgression may not exist or even be reversed. The second major result is that introgression should be impacted by how many diverged alleles an introgressing haplotype is replacing. Introgression of a haplotype that carries almost entirely new derived alleles is almost always more beneficial than a haplotype replacing existing derived alleles in the population (see Figs 2B and 3).

The predictions of our model generate several major expectations for patterns of introgression in nature that we cannot test here. First, large haplotypes that have managed to introgress should carry proportionately few ancestral or replacement alleles, as they should be strongly selected against otherwise. Second, even in the absence of asymmetric migration rates, there should be an asymmetry in the direction of introgression, with species that are more highly diverged in the pair showing less evidence of introgression. Haplotypes introgressing into a more diverged population are more likely to replace existing derived alleles, and so are more strongly selected against than haplotypes introgressing into a genetic background that has fewer substitutions in general. A rapidly adapting population therefore ends up with a genome that is more resilient to introgression than a sister population drifting around the ancestral optimum. Asymmetry in directionality of introgression has been

observed in many taxa, for example, *Mimulus*, *Anopheles*, and *Motacilla* [56–59]. It is often explained by Darwin's Corollary to Haldane's Rule [60]—asymmetry in postzygotic isolation resulting in lack of backcrosses to 1 direction of parents. Alternatively, as in our *D. melanogaster* data, the asymmetry may be driven by asymmetry in migration rates (with migrants from West Africa to North America, but not in reverse). Our model suggests that even in the absence of asymmetry of postzygotic isolation or migration, asymmetry of introgression can occur when one of the taxa has evolved more rapidly than the other. This same pattern may be mirrored along the genome, with regions of the genome which have many fixed substitutions being resilient to introgression. However, divergence at a single population is insufficient to generate an "island of speciation" [61–63], as introgression from the diverged population into a population carrying largely ancestral variants is still possible. Instead, islands of speciation that are resilient to ongoing gene flow between populations will only be generated when both populations fix multiple substitutions within the same region, creating barriers to introgression in both directions.

The current model shows how basic intuitions about the size of introgressing blocks are affected by epistatic fitness effects on the block of introgressing alleles and with the recipient genome. The results we obtain depend on several parameters that require far more study. Very few studies to date have examined the strength of epistatic interactions, and so parameterizing our model is still difficult. Similarly, very little is known about the dominance of epistasis. Lastly, our model, like any model, makes simplifying assumptions that are not likely to hold in many natural cases of introgression. Nonetheless, this work provides a novel framework to understand the fitness effects caused by the interplay between the size of introgressing allele blocks and divergence time.

## Materials and methods and model details

We model the fitness of an introgressing haplotype that carries $x$ alleles that are not present in the recipient population. For simplicity, we assume that the recipient population does not carry any polymorphisms aside from the introgressing haplotype. We will only examine the fitness of the introgressing haplotype while it is rare. As such, we assume that the average fitness of the population is determined entirely by the fitness of locally fixed alleles, relative to the fitness of a purely ancestral genome set to 1. The fitness of an individual carrying a set of derived alleles $X$ is defined to be:

$$W_X = \prod_{i=1}^{|X|}(1 + s_i) \prod_{i,j \in X, i \neq j}(1 + \varepsilon_{ij}) \tag{10}$$

Where $s_x$ is the direct selection coefficient of allele $x$ and $\varepsilon_{ij}$ is the epistatic effect between alleles $i$ and $j$. That is, we assume that the total fitness is a product of all fitness effects, although largely the same approximations are obtained assuming fitness is the sum of these effects. The fitness of an individual carrying the introgressing haplotype will therefore depend on the alleles the haplotype is carrying—these will be the diverged alleles outside the haplotype ($B \backslash X_B$, i.e., all diverged alleles in B minus those that are overwritten by the introgressing haplotype) and the new diverged alleles brought in by the haplotype (set $X_A$). A carrier of an introgressed haplotype therefore has the set of alleles I = $X_A \cup B \backslash X_B$. Finally, we assume that all fitness effects have low variance, so that total fitness is well approximated by using the average of these effects (e.g., $s_i = s$ and $\varepsilon_{ij} = \varepsilon_w$ or $\varepsilon_b$). This assumption may not be held in interaction networks that are highly structured (with some mutations having many strong interactions and others having few weak ones), causing high variance in the possible fitness effects. However, the expected fitness should not be affected (see [64,65]). The fitness of an individual carrying the

introgressed haplotype is therefore:

$$W_I = \prod_{i=X_A \cup (B \setminus X_B)} (1 + s_i) \prod_{i,j \in X_A, i \neq j} (1 + \varepsilon_{ij}) \prod_{i,j \in (B \setminus X_B), i \neq j} (1 + \varepsilon_{ij}) \prod_{i \in X_A, j \in (B \setminus X_B)} (1 + \varepsilon_{ij}) \quad (11)$$

Where $\varepsilon_w$ is the average epistatic effect for alleles fixed "within" the same population, while $\varepsilon_b$ is the epistatic effect for alleles fixed "between" populations. The relative selection coefficient can then be calculated as $W_I$ divided by the average fitness of an individual in B, given by the left side of Eq (1), minus 1. These equations are used to produce Fig 2. To obtain analytically simple results, we note that when the epistatic effects and direct selection are very small, the above is well approximated by:

$$W_I \approx 1 + s(x_A + b - x_B) + \varepsilon_w(x_A + (b - x_B)) + \varepsilon_b x_A(b - x_B) \quad (12)$$

Where $b = |B|$ or the size of set $B$. Note that alleles in sets $X_A$ and $X_A$ can overlap (i.e., a replacement allele is a locus where population B has a fixed substitution that also has a fixed substitution in population A). If we think of the 3 types of introgressing alleles outlined in Fig 1B, $x_A = x_{\text{new}} + x_{\text{replacement}}$, while $x_B = x_{\text{ancestral}} + x_{\text{replacement}}$. Early in the process of divergence, it is reasonable to assume that the number of replacement alleles will be negligible, and so to simplify results we assume that $x_{\text{replacement}} = 0$, but see Fig 3 for an exploration of how replacement alleles affect the fitness of the introgressing haplotype. We can next assume that $x_A = (1-f)x$ and $x_B = f x$, and plug in these values to obtain Eq (6). Finally, we can take the derivative of the resulting equation with respect to $x$ while setting $f = \frac{1}{2}$ to obtain Eq (9).

## Diploid model

We now need to consider the diploid case. We again assume the haplotype is quite rare, thus will always be found in a heterozygous state. As such, we need to consider 2 types of dominance effects—dominance of direct and epistatic effects. We assume direct effects have, on average, dominance $h$. Epistasis between 2 heterozygous-derived alleles will be assumed to be of strength $a_1\varepsilon$, while epistasis between a homozygous and heterozygous-derived pair of alleles is of strength $a_2\varepsilon$. Epistasis between 2 homozygous-derived alleles is of full strength $\varepsilon$. Like the haploid model, we can express the fitness of an individual carrying the introgressed haplotype as follows:

$$W_I = \prod_{i=X_A \cup X_B} (1 + hs) \prod_{i=B \setminus X_B} (1 + s)$$

$$\prod_{i,j \in X_A, i \neq j} (1 + a_1\varepsilon_w) \prod_{i,j \in X_B, i \neq j} (1 + a_1\varepsilon_w)$$

$$\prod_{i \in X_B, j \in (B \setminus X_B)} (1 + a_2\varepsilon_w) \prod_{i,j \in (B \setminus X_B), i \neq j} (1 + \varepsilon_w)$$

$$\prod_{i \in X_A, j \in X_B} (1 + a_1\varepsilon_b) \prod_{i \in X_A, j \in (B \setminus X_B)} (1 + a_2\varepsilon_b) \quad (13)$$

And the effective selection coefficient can be calculated by comparing this fitness to the fitness of an average individual in the population, still equal to Eq (1). These equations are used to produce S5 Fig. Following the same logic as before, we can approximate the relative selection

coefficient further by assuming all effects are small and have low variance to produce:

$$\sigma_I \approx sx(h-f) + \varepsilon_w\left(a_1\left(\frac{x(x-1)}{2} - x^2f(1-f)\right) + a_2xf(b-xf) + \frac{xf}{2}(xf+1) - bxf\right)$$
$$+ \varepsilon_b x(1-f)(a_1xf + a_2(b-xf)) \tag{14}$$

Finally, we can simplify the above in the special case that epistasis is additive ($a_1 = \frac{1}{2}\, a_2 = \frac{1}{4}$) to obtain a slightly condensed form:

$$\sigma_I \approx sx(h-f) + \frac{\varepsilon_w}{2}\left(\frac{1}{2}\binom{x}{2} + xf\left(1 - b - \frac{x}{2}(1-f)\right)\right) + \frac{\varepsilon_b}{2}x(1-f)\left(b - \frac{xf}{2}\right) \tag{15}$$

A few takeaways can be obtained from these equations. Just as in the haploid case, a linear relationship between the effective selection coefficient and divergence is expected. On the other hand, the size of the haplotype appears primarily in quadratic terms, and so large haplotypes at low divergence may once again be positively selected easily. Finally, we can make a note on dominance when $h = f$—for example, if direct selection is on average additive ($h = \frac{1}{2}$) and the introgressing haplotype carries new alleles as frequently as ancestral ($f = \frac{1}{2}$), no net direct fitness effects are expected and only epistatic effects determine selection on introgressing haplotypes.

### Re-analysis of *Drosophila melanogaster* data

We used previously calculated values of $f_D$ for introgression between fly lines with a majority ancestry West and those with majority ancestry OOA2 [37], assuming a population tree of (South1,(West,(OOA1,OOA2))). Individuals with less than 1% introgressed ancestry, but majority of either OOA1, OOA2, West, or South1 ancestry were identified, and $F_{ST}$ between these individuals was calculated using *pixy* [66] in windows matching the windows from introgression analyses. PBS [42] values for each branch were next calculated manually in R [67]. Windows with negative PBS values for any of the windows were next filtered out; however, major results are largely the same if these windows are kept (S6 Fig). Finally, branch scores were obtained by normalizing the PBS value at each window by dividing by the sum of the 3 PBS values. The resulting values were plotted in ternary plots using *ggplot* [68], *hexbin* [69], and *ggtern* [70] packages. Recombination rates per window were obtained using data from [71], assigning average recombination rate for each window based on the Comeron dataset. We next fit a general linear model of $f_D$ dependent on recombination and the OOA2 branch score using the *glm* function. While we could include other PBS branches as well, their high degree of correlation made us cautious of using more than 1 PBS branch in the model at a time.

### Supporting information

**S1 Fig. The effects of varying epistasis within and between populations.** Ternary plots as in Fig 3, with $b = 60$ and $x = 40$, $s = 0$. Selection on the haplotype varies widely depending on parameter choice. When between population interactions are positive (top 2 rows), introgression is favored in a wide range of scenarios, while deleterious between population interactions (bottom 2 rows) lead to small parameter spaces in which introgression may be favored. Epistasis within populations, on the other hand, has varying effects depending on the proportion of alleles which are replacement (R) or ancestral (A). When epistasis within and between are equal magnitude but opposite signs, introgression does not depend heavily on the mix of ancestral vs. replacement vs. novel alleles (diagonal from top left). The code underlying this figure can be located in S1 File.
(EPS)

**S2 Fig. A graphical illustration of Eq (9), demonstrating the size at which the introgressing haplotype is most strongly selected against versus the fraction of the haplotype that carries ancestral/replacement alleles.** The above shape is consistent for all values of parameters, and values are all positive when between and within population epistasis are opposite signs or are both positive. The code underlying this figure can be located in S1 File.
(EPS)

**S3 Fig. Replacement alleles show similar effects to ancestral.** Examining how the selection coefficient on an introgressing haplotype changes as the fraction of substitutions it carries varies between new, ancestral, and replacement alleles, as in Fig 3 but using approximation in Eq (5). The code underlying this figure can be located in S1 File.
(TIF)

**S4 Fig. Schematic representation of fitness effects in a diploid model.** Individuals carrying the introgressed haplotype are assumed to always be heterozygous, therefore carrying epistatic effects of varying degrees of dominance, with all new within population epistatic interactions occurring between heterozygous alleles, while some between population interactions occur between heterozygous and homozygous alleles. We parameterize epistatic dominance with 2 new parameters, $a_1$ and $a_2$.
(EPS)

**S5 Fig. Results for a diploid model.** All parameters are the same to those used in Fig 2, and only exact numeric solutions are shown. Three new parameters, $h$, $a_1$, and $a_2$ determine the dominance of direct selection and epistasis. (A) Varying $f$ for different values of $h$ demonstrates that increasing dominance decreases the impact of replacing existing variation. (B) When an individual is heterozygous at 2 derived alleles, epistasis is of strength $a_1{}^*\varepsilon$, when one is heterozygous and the other is homozygous, it is $a_2{}^*\varepsilon$, and when both are homozygous it is $\varepsilon$. Top row of plots shows how varying divergence ($b$) changes selection on introgressing haplotypes of varying sizes, $f$ fixed at 0. Bottom row shows the effects of varying the fraction of the haplotype that is ancestral alleles ($f$), $b$ fixed at 60. Patterns in general mirror the haploid case, although recessive epistasis leads to much stronger selection against the introgressing haplotype, and dominant epistasis leads to overall stronger selection for introgression. As with other models where $f$ is used, we assume that the number of replacement alleles is 0, but the shape of the curve changes very little with the addition of replacement alleles (as each replacement allele increases $x$ by only 1, but carries the effects of both a new and ancestral alleles). The code underlying this figure can be located in S1 File.
(EPS)

**S6 Fig. Raw PBS values, including negative ones plotted against $f_D$.** While both PBS values are negatively correlated with introgression, note that the vast majority of $PBS_{West}$ values are close to 0, and the negative trend is driven in part by more positive than negative PBS values. On the other hand, a more clear negative relationship between $PBS_{OOA2}$ and $f_D$ is observed. The code and data underlying this figure can be located in S1 File.
(EPS)

**S1 File. Compressed archive of files to regenerate figures in this manuscript.** Files include a Mathematica notebook to generate Figs 2 and 3, S1, S2, S3, and S5 Figs. The archive also includes data files and R code to generate Fig 4 and S6 Fig.
(ZIP)

## Acknowledgments

We would like to thank our reviewers, Adam Stuckert, Jenn Coughlan, Gaston Jofre, Jonathan Rader, Sean Anderson, and Tyler Kent for helpful comments.

## Author Contributions

**Conceptualization:** Andrius J. Dagilis, Daniel R. Matute.

**Funding acquisition:** Daniel R. Matute.

**Investigation:** Andrius J. Dagilis.

**Methodology:** Andrius J. Dagilis.

**Project administration:** Daniel R. Matute.

**Supervision:** Daniel R. Matute.

**Visualization:** Andrius J. Dagilis.

**Writing – original draft:** Andrius J. Dagilis, Daniel R. Matute.

**Writing – review & editing:** Andrius J. Dagilis, Daniel R. Matute.

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
