## [Editor Report · Decision Letter 0]

26 Apr 2023

Dear Dr Dagilis, 

Thank you for submitting your revised manuscript entitled "The fitness of an introgressing haplotype" for consideration as a Research Article by PLOS Biology.

Your revisions have now been evaluated by the PLOS Biology editorial staff, and I am writing to let you know that we would like to send your submission out for further assessment by the Academic Editor and possible re-review.

IMPORTANT: Please could you upload a marked-up "track changes" version of your manuscript as an additional file, indicating the differences between your new submission and the previous version, when you upload your addition metadata (see next paragraph)?

Once your full submission is complete, your paper will undergo a series of checks in preparation for re-review. After your manuscript has passed the checks it will be sent out for review. To provide the metadata for your submission, please Login to Editorial Manager (https://www.editorialmanager.com/pbiology) within two working days, i.e. by Apr 28 2023 11:59PM.

Kind regards,

Roli Roberts

Roland Roberts, PhD

Senior Editor

PLOS Biology

rroberts@plos.org

---

## [Decision Letter · Decision Letter 1]

18 May 2023

Dear Dr Dagilis,

Thank you for your patience while we considered your revised manuscript "The fitness of an introgressing haplotype" for publication as a Research Article at PLOS Biology. This revised version of your manuscript has been evaluated by the PLOS Biology editors, the Academic Editor and the original reviewers.

Based on the reviews, we are likely to accept this manuscript for publication, provided you satisfactorily address the remaining points raised by the reviewers and the following data and other policy-related requests.

IMPORTANT:

a) Please address the remaining requests from reviewers #2 and #3.

b) Is there any way of making the Title more informative? I quite like the almost philosophical tone of your current one, but it might be helpful to include the main findings (I couldn't think of a succinct way of doing this...).

c) Please address my Data Policy requests below; specifically, we need you to supply the numerical values underlying Figs 2AB, 3, 4BC, S1, S2, S3, S5ABCD, S6AB (and/or the code required to generate them), either as a supplementary data file or as a permanent DOI’d deposition.

d) Please cite the location of the data/code clearly in all relevant main and supplementary Figure legends, e.g. “The data underlying this Figure can be found in S1 Data” or “The data underlying this Figure can be found in https://doi.org/10.5281/zenodo.XXXXX”
https://osf.io/XXXXX

e) I note that you mention two of the reviewers (Barton, Fraisse) in the Acknowledgements. While we appreciate the sentiment, this is against PLOS policy, so please could you remove this?

We expect to receive your revised manuscript within two weeks. 

*Published Peer Review History*

*Press*

Sincerely,

Roli Roberts

Roland Roberts, PhD

Senior Editor,

rroberts@plos.org,

PLOS Biology

DATA POLICY:

Regardless of the method selected, please ensure that you provide the individual numerical values that underlie the summary data displayed in the following figure panels as they are essential for readers to assess your analysis and to reproduce it: Figs 2AB, 3, 4BC, S1, S2, S3, S5ABCD, S6AB. NOTE: the numerical data provided should include all replicates AND the way in which the plotted mean and errors were derived (it should not present only the mean/average values).

DATA NOT SHOWN?

REVIEWERS' COMMENTS:

Reviewer #1:

The authors have made extensive revisions to the manuscript text in response to a thorough set of reviewer comments. I believe these changes have greatly improved the manuscript's clarity, and in retrospect, the simulation study was probably an unreasonable ask. I have no further comments or concerns. 

Reviewer #2:

[identifies himself as Nick Barton]

This revision is a considerable improvement, and most responses are reasonable. However, some sections are still confusing, and need to be fixed before publication.

124 There are several problems with the very first paragraph in this section.  A "haplotype" is never defined, and it is not stated that recombination will be ignored.  It needs to be said right at the start that we consider a small non-recombining block, introgressing from a into b.

126 The # of substitutions in A, a, is never actually used.  Better to say at the start that all that matters are the effects of the alleles carried by the small block, which will interact with the b alleles in the recipient genome.

128 ""the marginal  fitness of a rare introgressing haplotype is (WI) compared to the average individual in the recipient population (® ), and calculate the effective selection coefficient () for the haplotype,  equal to WI/® -1" ([pdf](zotero://open-pdf/library/items/QHA4GQ2V?page=22))  What is meant here? Surely, we are comparing the marginal effect of the introgressing block with the marginal effect of the homologous block within the recipient. As it is written, we seem to be comparing the effects of a small block with the fitness of a whole individual - which is confusing.

The middle product term in Eq 1 needs to be written in the general form (with s_i, eps_w,i,j), as in Eq 8.  This can then be approximated by assuming an "average" s and eps_w.  Similarly in Eq 2: the general form should be stated clearly, and then approximated.

132 "all fitness effects are multiplicative" cannot be literally true, since this is a model of epistasis!

147 "fitness depends.. at least quadratically on epistatic effects". No! The formula is linear in epsilon. What is meant is "quadratically on the number of pairwise epistatic interactions"

170, 195 and 247: the order of magnitude difference must depend on assumptions that went into the simulations; this will not generally hold.

Eqs 5,6: There can be some further aproximation by noting that b>>1, and also that we expect b\\*eps to be of the same order as s (or rather, this is a natural scaling to make the contributions to fitness comparable). We then expect Eq 6 to further simplify to: "(1 − 2)() − (_ − _b(1 − ))(b)" 

222-224 I don't see that the quadratic term in x implies that the size of the haplotype has a larger effect than divergence, since b>>x surely?

Fig 3 - It is unreasonable to show such large values of x - which imply that the introgressing block is the same size as the whole genome (x~b).  This is perhaps less unreasonable for Drosophila, which has a weirdly small # of chromosomes, but  does not make much sense even there.

456-457 Again, the quadratic vs linear comparison is misleading since one expects x<<b.

672 effect-> affected

Eq 9 should be written with explicit subscripts s_i and eps_w,i,j etc

Reviewer #3:

[identifies herself as Christelle Fraïsse]

The authors made efforts to address most of the concerns raised during the reviewing process. The updated version of the manuscript includes new results, such as in the Supp. Fig. 1 (exploring a broader range of within and between epistasis values), and Supp. Fig. 5 (showing additional results for a diploid model). The authors also elaborated on several points that needed to be clearer in the first version, such as the variation of epistasis parameter values with divergence and consideration of genes network. They are also more cautious now with interpreting the Drosophila data (and I liked the new version of Figure 4 with densities depicted). Overall, this work is of high quality and relevant to our understanding of introgression patterns. It provides exciting predictions to be tested in a comparative genomics framework. Therefore I fully support its publication, although I still have a few minor suggestions below. Note that line numbers refer to the "tracked changes" version of the manuscript.

1) The authors do not consider the effect of recombination in their model in this updated version, which makes hard to draw conclusions on the relationship between recombination rate and introgression frequency. I understand that the authors failed to include recombination in their analytical model, and that running simulations would be cumbersome. They now briefly discuss how the effect of recombination may contribute to their model, which is OK. However, I'd suggest putting more emphasis on predictions about the direction and genomic location of introgression rather than on the relationship between recombination and introgression, especially in the abstract and Introduction.

2) The new version of Fig. 2 uses a lower value of direct fitness effects (i.e. s=-10^-2 instead of -10^-3 previously). This change renders the parameter space in which the positive interactions can cancel out the deleterious effects of the introgressing block much reduced compared to the previous version. Basically, the selection coefficient of the block is positive only when divergence is low (b=30) and the haplotype is large (x>25), which would mean that the haplotype introgressing carries nearly all the divergent sites. Moreover, the selection coefficient of the large haplotypes is similar to that of short ones, while in the previous version, it was much higher. Thereby, the prediction that the introgression of large haplotype can be favored in early divergence is strongly dependent on the underlying fitness landscapes (i.e. direct and epistatic selection coefficients). This could be discussed a bit more along lines 267 - 283.

3) Other minor points:

● L52 - 56: you list three patterns suggestive of the non-neutrality of introgressed regions. However, as it reads, it is unclear whether all patterns relate to humans or it is just the second one. Rephrasing would help to clarify.

● L70: consider replacing "without recombination" with "in the absence of recombination".

● L106: consider replacing "sex chromosomes" with "sex-limited chromosomes".

● L121: consider replacing "location" with "genomic location".

● L383: typo in "...general case of positive within and negative between [ ] this leads..."

● L745: typo in "...the expected fitness should not be [effect]..."

---

## [Editor Report · Decision Letter 2]

6 Jun 2023

Dear Andrius,

Thank you for the submission of your revised Research Article "The fitness of an introgressing haplotype changes over the course of divergence and depends on its size and genomic location" for publication in PLOS Biology. On behalf of my colleagues and the Academic Editor, Nick Barton, I'm pleased to say that we can in principle accept your manuscript for publication, provided you address any remaining formatting and reporting issues. These will be detailed in an email you should receive within 2-3 business days from our colleagues in the journal operations team; no action is required from you until then. Please note that we will not be able to formally accept your manuscript and schedule it for publication until you have completed any requested changes.

Sincerely, 

Roli

Senior Editor

PLOS Biology

rroberts@plos.org